# Effect of Surface Coating of Gold Nanoparticles on Cytotoxicity and Cell Cycle Progression

**DOI:** 10.3390/nano8121063

**Published:** 2018-12-17

**Authors:** Qian Li, Chun Huang, Liwei Liu, Rui Hu, Junle Qu

**Affiliations:** Key Laboratory of Optoelectronic Devices and Systems of Ministry of Education and Guangdong Province, College of Optoelectronic Engineering, Shenzhen University, Shenzhen 518060, China; liqian123@szu.edu.cn (Q.L.); huangchun1190@163.com (C.H.); liulw@szu.edu.cn (L.L.); jlqu@szu.edu.cn (J.Q.)

**Keywords:** cell cycle, nanoparticle location, surface biocompatibility, microtubule, proteomics

## Abstract

Gold nanoparticles (GNPs) are usually wrapped with biocompatible polymers in biomedical field, however, the effect of biocompatible polymers of gold nanoparticles on cellular responses are still not fully understood. In this study, GNPs with/without polymer wrapping were used as model probes for the investigation of cytotoxicity and cell cycle progression. Our results show that the bovine serum albumin (BSA) coated GNPs (BSA-GNPs) had been transported into lysosomes after endocytosis. The lysosomal accumulation had then led to increased binding between kinesin 5 and microtubules, enhanced microtubule stabilization, and eventually induced G_2_/M arrest through the regulation of cadherin 1. In contrast, the bare GNPs experienced lysosomal escape, resulting in microtubule damage and G_0_/G_1_ arrest through the regulation of proliferating cell nuclear antigen. Overall, our findings showed that both naked and BSA wrapped gold nanoparticles had cytotoxicity, however, they affected cell proliferation via different pathways. This will greatly help us to regulate cell responses for different biomedical applications.

## 1. Introduction

As the engineering of nanoparticles has been extensively developed over the past decades, various nanoparticles with unique physical and chemical properties have been designed for potential medical applications [1,2]. Improving our understanding of the interactions between nanoparticles and biological systems, especially at the cellular level, is crucial for their risk control and for evaluating their potential applications as drug delivery vehicles or therapeutic agents [3].The study of interactions between nanoparticles and biological systems, with an emphasis on elucidating the relationship between the physicochemical properties of nanoparticles and biological responses, is essential [4,5]. Such studies are important prerequisites for designing and engineering nanoparticles with intentionally enhanced or suppressed cellular responses and toxicity. However, the mechanisms mediating cellular responses to nanoparticles remain unclear, particularly about the effects of nanoparticles on cell cycle arrest at different phases.

As the cell cycle is closely related to cell proliferation and cytotoxicity, elucidating the mechanisms of different nanostructures on the regulation of cell cycle will be of great importance [6].Nanoparticles have been shown to cause cell cycle arrest, including G_2_/M and G_0_/G_1_ arrest. The type and extent of cell cycle arrest varies depending on the composition, size, size distribution, surface modification, and subsequent surface derivatization of nanoparticles [7,8,9]. G_0_/G_1_ arrest can be caused by DNA damage and microtubule damage, while nanoparticles in combination with oxidative stress and/or lysosome rupture could lead to G_0_/G_1_ arrest. However, the mechanisms and the factors behind the G_2_/M cell cycle arrest caused by nanoparticles are still unclear. Recently, Mahmoudi et al. speculated that the effects of nanoparticles on the cell cycle may depend on the intracellular location of the nanoparticles [6]. Additionally, Choudhury et al. reported that gold nanoparticles (GNPs) with lysosomal escape ability localized to the tubulin/microtubule system and caused cell cycle arrest at G_0_/G_1_ phase through induction of microtubule damage [10]. However, whether the intracellular localization of nanoparticles is linked with G_2_/M cell cycle arrest is still unknown.

GNPs have been recognized as promising nanoprobes in biomedical applications for clinical translation. Although they were once believed to be biocompatible, they are now known to cause cell cycle arrest and show unexpected toxicity to mammalian cells. In addition, the results of various studies have differed due to the use of GNPs with different physicochemical properties [10,11,12]. Thus, further studies are needed to evaluate the mechanisms through which GNPs cause cell cycle arrest. In this study, two types of GNPs with cetyltrimethylammonium bromide (CTAB)/bovine serum albumin (BSA) coatings were evaluated using raw 264.7 macrophage cells. Although there are earlier reports about BSA coated gold nanoparticles, their mechanism for modulating the cell cycle is still unknown. The correlations between biocompatibility, subcellular localization and cell cycle arrest of GNPs were investigated. We believe that our findings will improve the current understanding of the mechanisms behind which nanoparticles induce cell cycle arrest at different phases.

## 2. Materials and Methods

### 2.1. Materials

Hydrogen tetrachloroaurate (HAuCl_4_·3H_2_O), silver nitrate (AgNO_3_), CTAB, sodium salicylate, ascorbic acid, sodium citrate, poly(4-styrenesulfonic acid-co-maleic acid) sodium salt (PSSMA), and BSA were obtained from Sigma (Saint Louis, MO, USA). Rat monoclonal anti-α-tubulin antibodies conjugated with Alexa Fluor 647, rat monoclonal anti-α-tubulin antibodies and horseradish peroxidase (HRP)-labeled goat anti-rat IgG were purchased from Abcam (Cambridge, UK). Dulbecco’s modified Eagle’s medium (DMEM), fetal bovine serum (FBS), and phosphate-buffered saline (PBS) were purchased from Gibco (New York, NY, USA). The annexin V-fluorescein isothiocyanate (FITC) apoptosis detection kit, and propidium iodide (PI) cell cycle assay kit, phalloidin-FITC cytopainter, and 4′,6-diamidino-2-phenylindole (DAPI) were purchased from Beyotime (Shanghai, China). Lyso Tracker Green DND-26 was obtained from Invitrogen (Carlsbad, CA, USA).

### 2.2. Preparation and Characterization of GNPs

GNPs were prepared as previously described [13]. Briefly, 5 mL of 0.5 mM HAuCl_4_ was mixed with 5 mL of 0.2 M CTAB solution, and 1 mL of 6 mM NaBH_4_ was then added. The solution was stirred for 2 min and incubated for 30 min to prepare the seed solution. To prepare colorless growth solution, 9.0 g CTAB plus 0.8 g sodium salicylate were dissolved in 250 mL warm water. Next, 6 mL of 4 mM AgNO_3_ solution and 1 mL of 0.064 M ascorbic acid were added. Finally, 0.8 mL seed solution was injected into the growth solution, stirred for 30 s and left undisturbed at 30 °C for 12 h for GNP growth. To obtain CTAB-GNPs, the precipitates after centrifugation were re-dispersed in 10 mL distilled water. According to previous reports [14,15],the CTAB-GNPs were further coated with the polyelectrolyte PSSMA and BSA under stirring to obtain BSA-GNPs.

The morphology of the nanoparticles was observed by TEM at an accelerating voltage of 100 kV (JEOL Co.,Tokyo, Japan). The size stability of nanoparticles was determined with a Brooke Haven Nanosizer at 37 °C with GNPs in DMEM medium (pH 7.4) at the concentration of 15 pM. Ultraviolet-visible (UV-vis) measurements were carried out using a Shimadzu UV 2700 spectrophotometer (Kyoto, Japan).

### 2.3. Flow Cytometry Analysis of the Cell Cycle and Apoptosis

RAW264.7 cells were obtained from the Shanghai Institutes for Biological Sciences (Shanghai, China) and routinely cultured in DMEM supplemented with 10% FBS at 37 °C in a humidified atmosphere with 5% CO_2_ in air. Progression of cells through the cell cycle was examined by flow cytometry in RAW264.7 cells treated with GNPs. RAW264.7 cells were plated at 4 × 10^4^ cells/cm^2^ in six-well plates and grown for 18 h. After incubation of cells with GNPs (0–30 pM) for 2 h, the medium was replaced with DMEM, and the cells were incubated for an additional 0–14 h. The interaction time between cells and GNPs in this study refers to incubation with GNPs in DMEM for 2 h plus incubation with GNP-free DMEM for 0–14 h. The cells were fixed in 70% ethanol, and DNA was stained with PI in the presence of 40 mg/mL DNase-free RNase A for 30 min at 37 °C in the dark. Cell cycle analysis was performed using flow cytometry according to the manufacturer’s instructions. Early apoptotic cells were quantified by annexin V-FITC, whereas late apoptotic and necrotic cells were identified by PI staining. After treatment with GNPs, cells were trypsinized, harvested, washed with PBS, incubated with annexin V-FITC/PI for 15 min at room temperature in the dark, and analyzed on a flow cytometer (FACSCalibur BD, CA, USA).

### 2.4. Confocal Microscopy Analysis

Intracellular localization of GNPs was visualized using a Leica SP8 fluorescence confocal microscope (Wetzlar, Germany). RAW264.7 cells were treated with GNPs as described in the previous section and then co-incubated with Lyso Tracker Green DND-26 at 37 °C for 0.5 h. After washing with PBS, live cell imaging of green fluorescence with Lyso Tracker Green and diffusion reflection of GNPs irradiated at 630 nm were measured using a Leica SP8 (Wetzlar, Germany).

The morphology of the cytoskeleton was determined by immuno-staining RAW264.7 cells with a microtubule/microfilament fluorescent probe. Briefly, cells were fixed with 4% formaldehyde for 10 min, permeabilized with 0.1% Triton X-100 for 5 min, blocked with 1% BSA for 1 h, and then incubated with anti-tubulin antibodies conjugated with Alexa Fluor 647 and/or phalloidin-FITC according to the manufacturer’s instructions for staining microfilament and microtubules. The cells were co-incubated with DAPI for nuclear staining. Polychromatic images of cells were measured using a Leica SP8 for green fluorescence indicating microfilament, red fluorescence indicating microtubules, and blue fluorescence indicating nuclei. Diffusion reflection of GNPs was defined as yellow, green, or red to avoid confusion with fluorescence.

### 2.5. Western Blot Analysis

Changes in the ratios of free tubulin within the polymerized microtubules were measured by western blotting. After treatment with GNPs, cells were lysed with 0.2 mL lysis buffer (20 mM Tris-HCl (pH 6.8), 0.5% NP-40, 1 mM MgCl_2_, 2 mM EGTA, 1 mM orthovanadate, and 20 mg/mL aprotinin, leupeptin, and pepstatin). Centrifugation yielded soluble tubulin dimers in the supernatant and polymerized microtubules in the pellet. Pellets were solubilized with sodium dodecyl sulfate (SDS) lysis buffer (Beyotime; Shanghai, China). Cell lysates containing equal amounts of protein were separated by SDS-polyacrylamide gel electrophoresis, transferred, probed with specific antibodies against α-tubulin, and detected on X-ray films using the chemiluminescence technique.

### 2.6. qRT-PCR

RAW264.7 cells were incubated with GNPs and then harvested for examination of gene expression by qRT-PCR. Briefly, total RNA was extracted with TRIzol reagent (CW0580S; CWBIO, Beijing, China). cDNA was then synthesized using a SuperScriptFirst-Strand Synthesis kit (CW2569M; CWBIO). The specific primers used in this study were as follows: *p53*, (forward) 5′-GCTCCTCCCCAGCATCTTA-3′ and (reverse) 5′-GGGCAGTTCAGGGCAAA-3′; kinesin 5A, (forward) 5′-GGCGGAGACTAACAACGAA-3′ and (reverse) 5′-CTTGGAAAATGGGGATGAA-3′; glyceraldehyde 3-phosphate dehydrogenase, (forward) 5′-AAGAAGGTGGTGAAGCAGG-3′ and (reverse) 5′-GAAGGTGGAAGAGTGGGAGT-3′. The relative expression of mRNA was calculated by the 2^−−ΔΔct^ method.

### 2.7. Proteomics Analysis

Cells were incubated with GNPs (30 pM) for 6 h, and cellular protein was extracted, digested, and desalted. The resulting peptide mixture was labeled with an iTRAQ Reagent-8 plex Multiplex Kit (AB Sciex U.K. Limited, Sheffield, U.K.) according to the manufacturer’s instructions. Next, the labeled samples were fractionated using high-performance liquid chromatography (Thermo DINOEX Ultimate 3000 BioRS, Waltham, USA) using a Durashell C18 column (5 μm, 100 Å, 4.6 × 250 mm, Tianjin, China). Liquid chromatography electrospray ionization tandem mass spectrometry (MS/MS) analysis was performed on an AB SCIEX nanoLC-MS/MS (Triple TOF 5600 plus) system. Briefly, samples were chromategraphed on a C18 column (3 µm, 75 µm × 150 mm) with a 90-min gradient elution. Buffer A (0.1% formic acid and 5% acetonitrile) and buffer B (0.1% formic acid and 95% acetonitrile) served as the mobile phase. MS1 spectra were collected in the range of m/z 350–1500 for 250 ms, and the 30 most intense precursor ions were selected for fragmentation. MS2 spectra were collected in the range of m/z 100–1500 for 50 ms. Precursor ions were excluded from reselection for 15 s. The original MS/MS file data were submitted to ProteinPilot Software v4.5 (AB Sciex Pte Ltd. Waltham, MA, USA; https://sciex.com/products/software/proteinpilot-software) for data analysis. For protein identification, Paragon algorithm2, which was integrated into ProteinPilot, was employed against Uniprot Mus musculus 20171124.fasta (84434 items, updated in November 2017) for database searching. The parameters were set as follows: Instrument, TripleTOF 5600; iTRAQ quantification; cysteine modified with iodoacetamide. The biological modifications included ID focus and trypsin digestion, and the quantitate, bias correction, and background correction was used for protein quantification and normalization. Only proteins with at least one unique peptide and an unused value of more than 1.3 were considered for further analysis.

## 3. Results

### 3.1. Proertiesof GNPs withBSA/CTABCapping Agents

As the surfactants have poor biocompatibility, several shells such as carbon shells and biopolymer shells have been used to reduce toxicity of surfactants [16]. We chose BSA as a model molecule for its biocompatibility. Zeta potential of CTAB-GNPs and BSA-GNPs are 28.4 ± 2.6 mV and −20.5 ± 2.1 mV respectively, indicating positive CTAB and negative BSA on the surface of GNPs. The physicochemical properties of BSA-GNPs and CTAB-GNPs are shown in Figure 1. Transmission electron microscopy (TEM) showed that both types of nanoparticles were rod shaped with similar particle sizes in distilled water (Figure 1A,B). Although the hydrodynamic diameter deduced from the Stokes-Einstein equation was not accurate when regarding nano-rods as nano-spheres, the diffusion coefficient determined by dynamic light scattering (DLS) was still accurate. The particle size peak can be a signature for determining the nano-rod aggregation formation [17]. DLS analysis showed that the particle size of CTAB-GNPs increased from 45 to 79 nm as the incubation time increased. In contrast, no significant changes in particle size were observed for BSA-GNPs (Figure 1C). These results indicated coating with BSA enhanced the stability of GNPs, which can be attributed to good dispersity of BSA in high salt solution.The extinction spectra of the nanoparticles showed a peak at around 630 nm, which corresponded to the longitudinal surface plasmon resonance of the rod-shaped GNPs (Figure 1D).

### 3.2. Effects of GNPs on the Cell Cycle and Apoptosis

Murine macrophages RAW264.7 were used owing to their strong nanoparticle phagocytosis and short cell cycle period. Apoptosis assays revealed that incubation of RAW264.7 cells with 15 pM of BSA-GNPs yielded 38.82% ± 4.30% early apoptotic cells and 33.98% ± 4.37% late apoptotic cells, whereas incubation of cells with 15 pM of CTAB-GNPs yielded 59.72% ± 1.52% early apoptotic cells and 10.23% ± 1.57% late apoptotic cells (Figure 2A,C). The apoptosis rate increased as the concentration of GNPs increased. Thus, for subsequent cell cycle analyses, we chose a dosage of 15 pM. Compared with the control group (7.71% ± 1.64% in G_2_/M phase), 18.54% ± 1.40% of cells were in the G_2_/M phase after treatment with 15 pM BSA-GNPs for 2 h. This indicated that BSA-GNPs induced cell cycle arrest at G_2_/M phase. Notably, for the cells treated with 15 pM CTAB-GNPs for 2 h, 62.88% ± 3.01% of cells were found to be in the G_0_/G_1_ phase, compared with 48.56% ± 1.57% in the control group. BSA-GNPs and CTAB-GNPs also induced G_2_/M and G_0_/G_1_ arrest after 16 h of treatment (Figure 2B,D), respectively.

### 3.3. Intracellular Localization of GNPs

Lysosomes, sliding on microtubules, play important roles in the intracellular transportation of nanoparticles [18]. As microtubules greatly affect the cell cycle, interactions of GNPs with lysosomes/microtubules were investigated. Figure 3 shows fluorescent images of GNP-treated cells in which the cell nucleus, lysosomes, microtubules, and GNPs were labeled in different channels. As shown in Figure 3A, the green fluorescence from Lyso Tracker Green DND-26 disappeared in most of the regions of the cells treated with CTAB-GNPs, indicating the disruption of lysosomes by CTAB-GNPs. This could be attributed to the surfactant CTAB, which facilitates lysosome escapees reported in previous reports [19]. However, different results were found in BSA-GNP-treated cells. The colocalization of green fluorescence from lysosomes and scattering reflection from BSA-GNPs (in red) showed that BSA-GNPs were accumulated in lysosomes. In addition, the accumulated green fluorescence of the Lyso Tracker Green DND-26 indicated an accumulation of lysosomes.

Figure 3B,C show the cytoskeleton morphology upon GNP treatment, as determined by laser confocal microscopy. Compared with the PBS treated cells, CTAB-GNPs caused shrinkage of microtubules, microfilaments, and nuclei after 2 h of nanoparticle treatment [20]. Additionally, CTAB-GNPs were aggregated into small circulars with diameters between 0.6–0.9 µm, and they were matched with the red fluorescence from α-tubulin after 16 h of treatment. These suggest that CTAB-GNPs induced tubulin aggregation. In contrast, BSA-GNPs treated cells showed increased microtubule and nuclear organization in the mitosis phase [21]. No overlap between the BSA-GNPs and microtubule-tubulin system was observed after 16 h of nanoparticle treatment.

### 3.4. Effects of Nanoparticles Ondepolymerization/Polymerization of Microtubules

To investigate the potential effects of nanoparticles on the depolymerization/polymerization of microtubules, western blotting was performed after separating the free tubulin from polymerized microtubules. In these cells, treatment with BSA-GNPs increased polymerized microtubules compared with the untreated control cells (Figure 3D), suggesting microtubule stabilization and inhibition of microtubule depolymerization. However, cells treated with CTAB-GNPs showed increased free tubulin, which may be due to the inhibition of microtubule polymerization and assembly of tubulin into small aggregates [10,22].

### 3.5. Protein Identification and Quantification by Quantitative Real-Time Reverse Transcription Polymerase Chain Reaction (qRT-PCR)

Kinesin 5A is a microtubule motor protein associated with lysosomes and acts as a microtubule polymerase by promoting tubulin polymerization and inhibition of tubulin depolymerization [23,24]. Compared with the control group, the mRNA level of kinesin 5A increased 1.26- and 1.91-fold in CTAB-GNP and BSA-GNP treated cells, respectively (Figure 3E). The increase in kinesin 5A could be attributed to the accumulation of lysosomes on microtubule during GNP transport. However, kinesin 5A levels in CTAB-GNP treated cells were much lower than those in BSA-GNP-treated cells, potentially because of the subsequent lysosome rupture induced by CTAB-GNPs. Overall, the significant increase of kinesin 5A (*p* < 0.01) suggested lysosome accumulation on microtubules and microtubule stabilization in BSA-GNP treated cells. As shown in Figure 3E, p53 mRNA levels were decreased by 8.92% in BSA-GNP-treated cells, yet increased by 1.21fold (*p* < 0.05) in CTAB-GNP treated cells. The increase of P53 can be contributed to microtubule disruption [25].

### 3.6. Protein Identification and Quantification by Isobaric Tags for Relative and Absolute Quantitation (iTRAQ)

To further explore the cell cycle arrest mechanism induced by GNPs, we used iTRAQ proteomics to identify and quantify protein changes in RAW264.7 cells before and after GNP treatment. In this study, 3341 and 3348distinct proteins were identified using iTRAQ-based proteomic technology in BSA-GNP and CTAB-GNP treatment, respectively (Appendix A). To improve our understanding of the roles of these proteins, differentially accumulated protein analysis was based on the fold-change >1.5 or <0.667 (*p* < 0.05). For the cells treated with BSA-GNPs, 159 proteins were found to be differentially expressed compared with the control, including 65 up-regulated and 94 down-regulated proteins. Moreover, 102 proteins were found to be differentially expressed in CTAB-GNP treated cells, including 55 up-regulated and 47 down-regulated proteins. As shown in the Venn diagram, 36 differentially expressed proteins were common in both GNP-treated groups (Appendix A). Gene ontology (GO) classification of these differentially expressed proteins were divided into three classes (biological processes, cellular components, and molecular functions). Cells treated with BSA-GNPs or CTAB-GNPs have shown differences in all the three classes (Appendix A).

### 3.7. Effects of Nanoparticles on Cell Cycle-Related Protein Expression

Kyoto Encyclopedia of Genes Genomes (KEGG) annotation analysis of all differentially expressed proteins was used to explore the underlying pathways and processes, and the top 10 altered pathways are shown in Appendix A. Down-regulation of cell cycle-related proteins was observed following BSA-GNP treatment. Down-regulation of actin cytoskeleton-related proteins, which are closely related to the cell cycle, were observed following CTAB-GNP treatment. We adopted KEGG annotation analysis to explore the underlying pathways of the cell cycle (Figure 4). Our results showed that three of these unique proteins (cadherin 1 (Cdh1), minichromosome maintenance complex component 5 (MCM5), 14-3-3 protein) were related to the cell cycle in BSA-GNP-treated cells. Of these proteins, the expression of Cdh1 increased 2.22 fold in response to mispositioned spindles. Cdh1 is an antagonist of the spindle assembly checkpoint and its over-expression could lead to the silencing of mitotic cyclin-dependent kinase 1 (CDK1) activity and consequently the cell cycle arrest at G_2_/M phase. MCM5, which was up-regulated in the transition from the G_0_ to G_1_/S phase of the cell cycle [26], was decreased 0.59 fold. Therefore, the reduction of MCM5 is implicated in low numbers of cells in the G_0_/G_1_ and S phases. The 14-3-3 protein zeta/delta, 14-3-3 protein gamma, and 14-3-3 protein tau were down-regulated 0.36–0.57 fold. The 14-3-3 protein directly binds to kinesin heterodimers and acts as a phospho-Ser/Thr-binding factor [27]. Phosphorylation of kinesin 5A inhibits its binding to microtubules [28]. Thus, we conclude that the down-regulation of 14-3-3 has weakened the phosphorylation of kinesin 5A and thus promoted the binding of kinesin 5Ato spindle microtubules. As a result, the microtubule was stabilized by BSA-GNPs. In CTAB-GNPtreated cells, proliferating cell nuclear antigen (PCNA) protein was up-regulated by 1.52 fold as compared with that in the control group. PCNA, as an accessory factor for DNA polymerases, is up-regulated rapidly in the G_1_phase through early S phase and is then down-regulated in late S and G_2_/M phases. Increased levels of PCNA can cause cell cycle arrest in G_0_/G_1_ through the inactivation of CDK4/6. Moreover, increased levels of p53 and PCNA can contribute to microtubule damage [25,29].

## 4. Discussion

In this study, we found that GNPs causing cell cycle arrest was dependent on biocompatibility of GNP surfaces. Coating of GNPs with biocompatible BSA induced G_2_/M arrest through microtubule stabilization, while residual toxic CTAB on the surface of GNPs typically caused cell cycle arrest in G_0_/G_1_ phase microtubule disruption. Kim et al. have shown that the cell cycle affects the intracellular transport of nanoparticles [30]. Nanoparticles internalized by cells are not exported from cells but are split during G_2_/M. Indeed, we found that the intracellular transport/location of nanoparticles had an effect on cell cycle progression (Figure 5). The accumulation of BSA-GNPs in lysosomes increased the level of kinesin 5A and caused subsequent stabilization of microtubules (including the promotion of tubulin polymerization and inhibition of tubulin depolymerization) [23,24], blockage of chromosome segregation, and induction of cell cycle arrest in G_2_/M viaCdh1 elevation [31]. In contrast, CTAB on the surface of GNPs caused lysosome/endosome rupture and subsequent microtubule damage through tubulin aggregation and the inhibition of tubulin polymerization. These changes induced G_0_/G_1_ arrest through the regulation of p53 and PCNA. Overall, biocompatibility properties of GNPs plays an important role in cell cycle progression. Biocompatible coated GNPs could inhibit lysosome rupture caused by residual surfactant and switched G_0_/G_1_ arrest to G_2_/M arrest. Similar results are expected when using other biocompatible molecule coated GNPs, including polyethylene glycol and fluoro-6-deoxy-d-glucose, in accordance with previous reports [32,33].

Microtubules are the major components of cytoskeletal systems that are responsible for regulation of the cell cycle. Many commonly used drugs, including paclitaxel (a microtubule-stabilizing agent), nocodazole (a microtubule-destabilizing agent), and vinblastine (a microtubule-destabilizing agent) induce G_2_/M cell cycle arrest through regulation of microtubules. Choudhury et al. reported that bare GNPs induce G_0_/G_1_ arrest by causing microtubule damage [10]. In this study, we demonstrated that BSA-coated GNPs stabilized microtubules and caused G_2_/M arrest by inducing interactions between lysosomes and microtubules. Nanoparticles are taken up and transported within subcellular structures that are surrounded by one or two layers of membranes, including endosomes, lysosomes, mitochondria, and multivesicular bodies [34]. The motility of these subcellular structures is based on microtubules. Therefore, the transport of nanoparticles can affect dynamic changes in microtubules. Microtubule-interfering drugs affect the cell cycle distribution by impairing the mitotic checkpoint and regulating the activity of cyclins and CDKs. Both stabilization and destabilization of microtubules could impair the mitotic checkpoint and cause G_2_/M arrest. For example, microtubule-stabilizing drug paclitaxel regulates the mitotic checkpoint proteins Bub1, CDK1 and CDK2 [35,36]. Microtubule-destabilizing drug nocodazole caused mitotic slippage through precocious activation of Cdh1 and inhibition of CDK1 [37,38]. BSA-GNPs in our study regulated CDK1 through up-regulation of Cdh1, so BSA-GNPs stabilizing microtubules may also lead to a potential cancer therapy.

## 5. Conclusions

GNPs causing cell cycle arrest was highly dependent on the surface biocompatibility of GNPs. Residual toxic CTAB on the naked GNPs typically caused cell cycle arrest in G_0_/G_1_ phase, whereas the coating of GNPs with BSA resulted in the inhibition of lysosome rupture ability, microtubule stabilization, and a switch to G_2_/M arrest. This will greatly help us to regulate the cell cycle progression through modulating surface coating and biocompatibility of nanoparticles and direct us to set the guidelines for the formulation of nanoparticles in different biomedical applications.More importantly, BSA-GNPs caused G_2_/M arrest through microtubule stabilization similarly to the mechanisms of many well-known anticancer drugs, and the recognition of this mechanism could be applied as a new therapeutic target of nanoparticles in tumor therapy.

## Figures and Tables

**Figure 1 nanomaterials-08-01063-f001:**
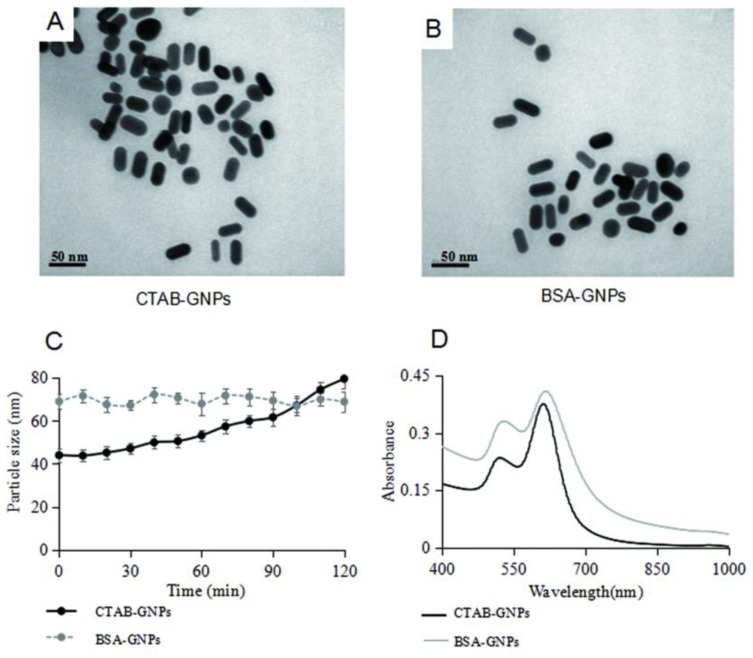
Characterization of GNPs. (**A**) TEM image of CTAB-GNPs in distilled water. (**B**) TEM image of BSA-GNPs in distilled water. (**C**) Hydrodynamic size of GNPs in DMEM. (**D**) UV-vis spectra of GNPs in distilled water.

**Figure 2 nanomaterials-08-01063-f002:**
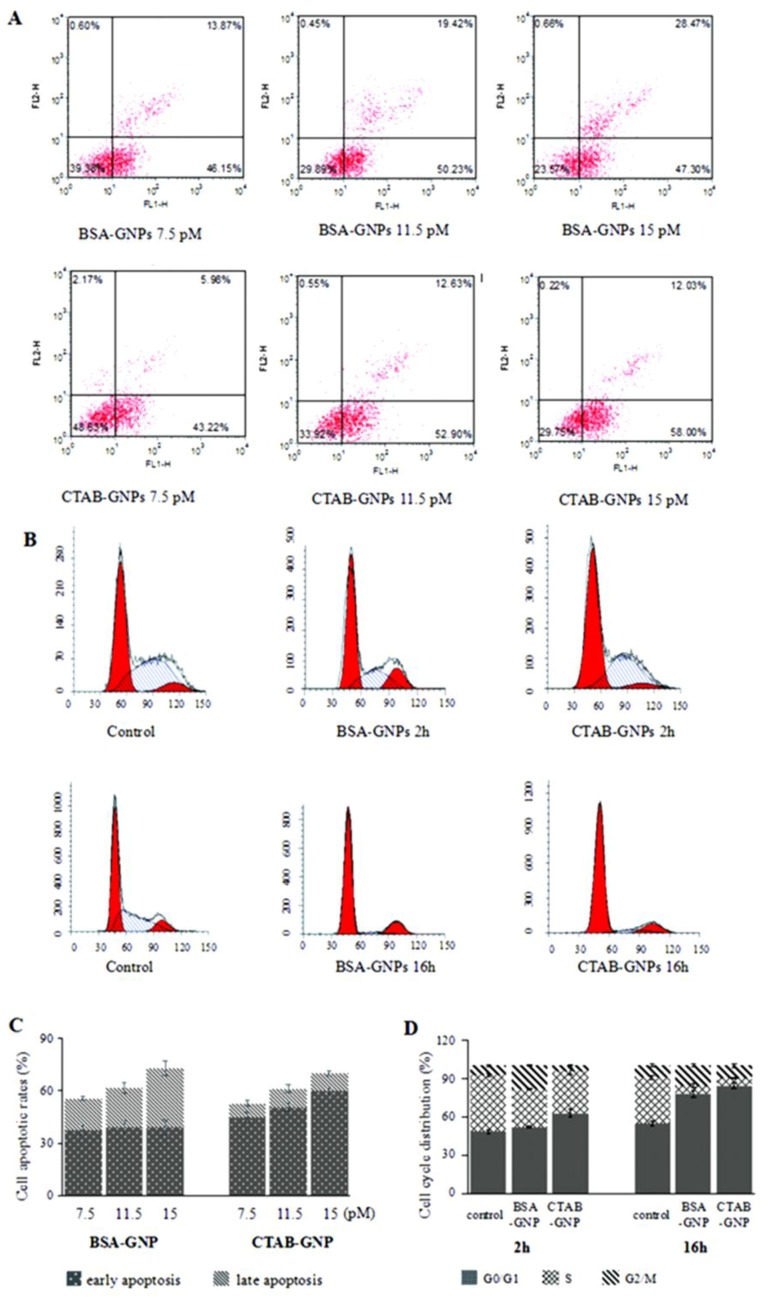
Apoptosis and cell cycle distributions of RAW264.7 cells before and after GNP treatment, with untreated cells used as control. (**A**) Flow cytometry images of GNPs inducing cell apoptosis after incubation for 16 h. (**B**) Flow cytometry images of cell cycle arrest in RAW264.7 cells treated with 15 pM BSA-GNPs and CTAB-GNPs for 2 or 16 h. (**C**) Barchart showing intensity of cell apoptosis. (**D**) Barchart showing distributions of the cell cycle.

**Figure 3 nanomaterials-08-01063-f003:**
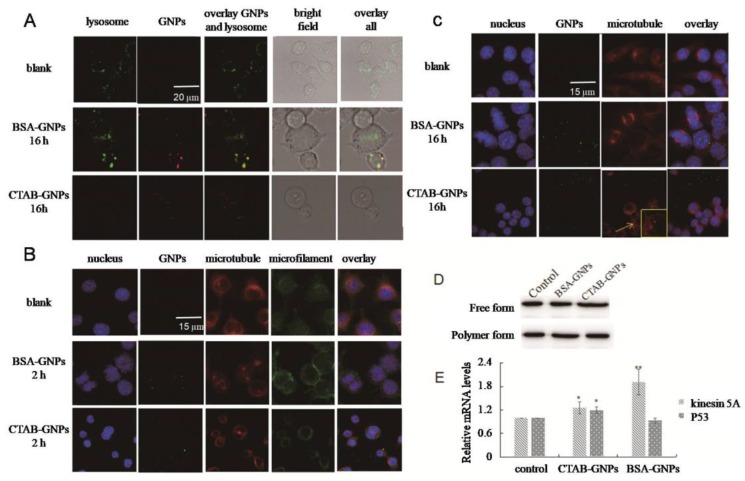
Effect of GNPs on subcellular organelles, with untreated cells used as control. (**A**) Confocal microscopy images of cell lysosome after incubation with GNPs for 16 h, showing colocalization of BSA-GNPs (red) with lysosomes (green). (**B**) Fluorescence microscopy images of cell cytoskeleton after incubation with GNP for 2 h, showing shrinkage of microtubules (red) and microfilaments (green) in CTAB-GNPs and increased microtubules (red) and nuclear (blue) organization in the mitosis phase in BSA-GNPs. (**C**) Fluorescence microscopy images of cell cytoskeleton after incubation with GNP for 16 h, showing colocalization of GNPs (green) with microtubules (red). (**D**) Western blot analysis of free tubulin and polymerized microtubule in cells treated with GNPs. (**E**) Relative mRNA levels of kinesin 5A and P53 in cells treated with GNPs.

**Figure 4 nanomaterials-08-01063-f004:**
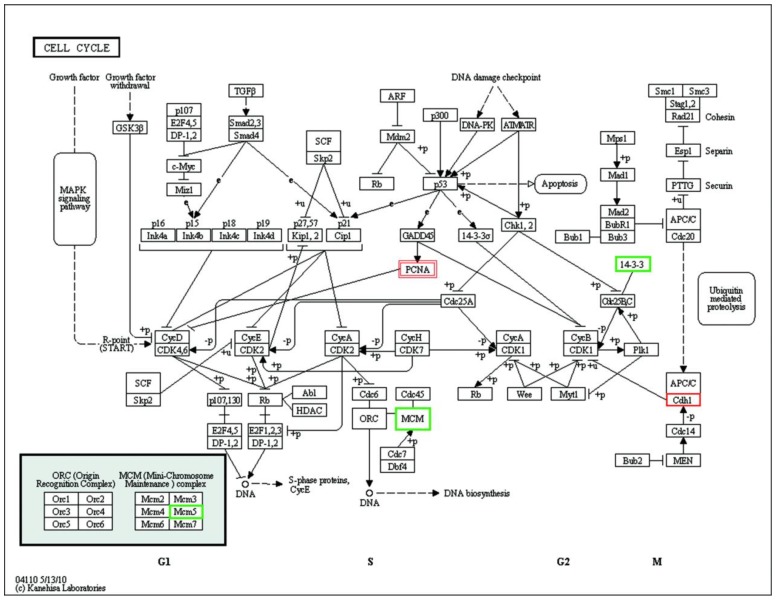
KEGG pathway analysis of the cell cycle in GNP-treated cells. PCNA was up-regulated in CTAB-GNP treated cells. Cdh1 was up-regulated, whereas 14-3-3 protein and MCM5 were down-regulated in BSA-GNP-treated cells. Single line frames refer to BSA-GNPs, and double line frames refer to CTAB-GNPs. Red frames indicate up-regulated proteins, and green frames indicate down-regulated proteins.

**Figure 5 nanomaterials-08-01063-f005:**
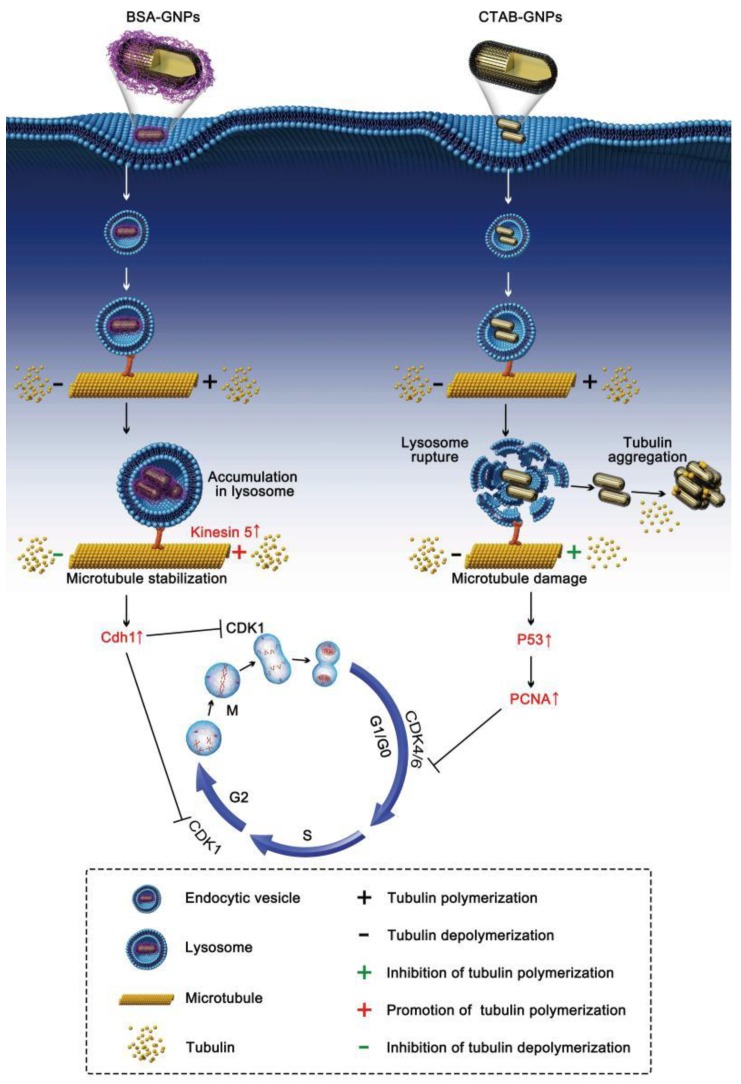
Mechanism through which GNPs causes cell cycle arrest was dependent on the biocompatible property of GNP surface. Coating of GNPs with biocompatible molecules, such as BSA, inhibited lysosome rupture and switched G_0_/G_1_arrest to G_2_/M arrest. The accumulation of BSA-GNPs in lysosomes increased the level of kinesin 5A and caused subsequent stabilization of microtubules (including promotion of tubulin polymerization and inhibition of tubulin depolymerization), blockage of chromosome segregation, and induction of cell cycle arrest in G_2_/M via Cdh1 elevation. In contrast, toxic CTAB on the surface of GNPs caused lysosome rupture and ssubsequent microtubule damage through tubulin aggregation. These changes induced G_0_/G_1_ arrest through regulation of p53 and PCNA.

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
