# Peer review of "Effect of Surface Coating of Gold Nanoparticles on Cytotoxicity and Cell Cycle Progression"

_nanomaterials, 2018, doi:10.3390/nano8121063_

Reviewer 1 Report
In this manuscript, gold nanoparticles were synthesized using the bovine serum albumin as coating agent. Synthesized nanoparticles were characterized by TEM, including the hydrodynamic size by using the zetasizer. Authors mentioned about the stability of the nanoparticles, but did the authors measure the zeta potential?. These newly synthesized gold nanoparticles were evaluated for the cyotoxicity of RAW cells. In addition to cell cycle progression, as well as the protein expression was also studied. But, this study lacks the rational and novelty of their presented results, as there are earlier reports about the BSA coated gold nanoparticles. Authors need to include the discussion about previous studies. It is deficient in several ways and does not warrant publication in its current form and the overall writing of the manuscript is very poor with too many grammatical mistakes. Manuscript needs to be thoroughly proof read by all the authors before the resubmission. For, example last line of the abstract was repeated twice.
Author Response
Thank you for valuable advices that have helped us to improve the manuscript. Changes are listed as follows.
1.Authors mentioned about the stability of the nanoparticles, but did the authors measure the zeta potential?
Zeta potential of CTAB-GNPs and BSA-GNPs are 28.4±2.6 mV and -20.5±2.1 mV respectively, indicating positive CTAB and negative BSA on the surface of GNPs. ” was added.
2. These newly synthesized gold nanoparticles were evaluated for the cyotoxicity of RAW cells. In addition to cell cycle progression, as well as the protein expression was also studied. But, this study lacks the rational and novelty of their presented results, as there are earlier reports about the BSA coated gold nanoparticles. Authors need to include the discussion about previous studies.
“Although there are earlier reports about the BSA coated gold nanoparticles, however, their mechanism for modulate cell cycle is still unknown. ” is added.
The novelties of our report have been marked red in lines 46-53. The novelties are as follows: “However,the mechanisms and the factors behind the G2/M cell cycle arrest caused by nanoparticles are still unclear.” and “However, whether the intracellular localization of nanoparticles is linked with G2/M cell cycle arrest is still unknown.”
3. It is deficient in several ways and does not warrant publication in its current form and the overall writing of the manuscript is very poor with too many grammatical mistakes. Manuscript needs to be thoroughly proof read by all the authors before the resubmission. For, example last line of the abstract was repeated twice.
The last line of the abstract has been deleted, and the grammatical mistakes have been corrected and marked red in the manuscript.
Reviewer 2 Report
In the opinion of this reviewer the manuscript, Nanomaterials-404691, is organized well and covers a topic that is worthy of investigation. The test nanomaterials are well characterized and their electron microscopic images are provided for the benefit of readers. This is an important contribution for the developing field of nanotoxicology. This reviewer would like to see this study presented as a hypothesis-driven one and, therefore, the hypothesis should be clearly defined up front and justified in the discussion. There are a few grammatical and typographical errors that the authors should address.
The authors state that “Overall, our findings have shown that both naked and BSA wrapped gold nanoparticles had cytotoxicity, however, they affected cell proliferation via different pathways. This will greatly help us to regulate cell response for different biomedical applications. This will greatly help us to regulate cell response for different biomedical applications.” However, in this reviewer’s opinion, they have not used all the appropriate sophisticated systems and biomarkers to draw this conclusion.
A thorough literature search indicates that the idea presented in this manuscript is not novel. For example, a similar article published by Yang et al two years ago [ Yang et al. 2016. Surface properties of plasma-functionalized graphite-encapsulated gold nanoparticles prepared by a direct current arc discharge method. J. Phys. D: Appl. Phys. 49 (2016) 185304 (11pp), doi: 10.1088/0022-3727/49/18/185304] presented such ideas. The article published by Yang et al is not even mentioned in the current manuscript being considered for publication in the “Nanomaterials”. This reviewer doesn't have an overall sense of the contribution that this manuscript is making to move the science in this area forward.
In conclusion, in the opinion of this reviewer, the manuscript is written in a publishable manner and the data is organized and presented well. However, the overall contribution to moving the science forward in this area seems minimal.
Author Response
Thank you for valuable advices that have helped us to improve the manuscript. Changes are listed as follows.
1. The authors state that “Overall, our findings have shown that both naked and BSA wrapped gold nanoparticles had cytotoxicity, however, they affected cell proliferation via different pathways. This will greatly help us to regulate cell response for different biomedical applications. This will greatly help us to regulate cell response for different biomedical applications.” However, in this reviewer’s opinion, they have not used all the appropriate sophisticated systems and biomarkers to draw this conclusion.
Thank your for your valuable advice, this article is a preliminary study, we will adopt your advice in our following study.
2. A thorough literature search indicates that the idea presented in this manuscript is not novel. For example, a similar article published by Yang et al two years ago [ Yang et al. 2016. Surface properties of plasma-functionalized graphite-encapsulated gold nanoparticles prepared by a direct current arc discharge method. J. Phys. D: Appl. Phys. 49 (2016) 185304 (11pp), doi: 10.1088/0022-3727/49/18/185304] presented such ideas. The article published by Yang et al is not even mentioned in the current manuscript being considered for publication in the “Nanomaterials”.
“As the surfactant have poor biocompatible, several shells can be used to reduce toxicity of surfactants such as carbon shells and biopolymer shells [14]. We chose BSA as model molecule for its important role in encapsulation. ” is added
“[14] Yang E, Chou H, Tsumura S, Nagatsu M. Surface properties of plasma-functionalized graphite-encapsulated gold nanoparticles prepared by a direct current arc discharge method. J. Phys. D: Appl. Phys. 2016, 49, 185304.” is added.
Reviewer 3 Report
This study explores gold nanoparticle coating type effects on cell cycle progress. The introduction clearly states the goal and provides adequate background information. The experimental part though lacks some critical details. First of all, the gold nanoparticles, used in the study have not been adequately characterized. The coating composition is largely assumed from the preparation chemistry. Authors would need to provide proof of coating composition for both gold particle types used as it is critical for their data interpretation. In particular, how the dynamic light scattering data was obtained, what are the charges (zeta potential), coverage by surface functionalities, etc. Also, please define the controls used in Figures 2 and 3. CTAB coated particles are referred as "toxic" although no data is provided. Could it be that CTAB detaches during incubations with cells and is responsible for the observed effects?
Author Response
Thank you for valuable advices that have helped us to improve the manuscript.Changes are listed as follows.
1. First of all, the gold nanoparticles, used in the study have not been adequately characterized. The coating composition is largely assumed from the preparation chemistry. Authors would need to provide proof of coating composition for both gold particle types used as it is critical for their data interpretation.
“Zeta potential of CTAB-GNPs and BSA-GNPs are 28.4±2.6 mV and -20.5±2.1 mV respectively, indicating positive CTAB and negative BSA on the surface of GNPs.” was added in line 177.
The preparation procedure had been reported previously, and “According to previous reports [14,15], was added in line 87.
2. In particular, how the dynamic light scattering data was obtained, what are the charges (zeta potential), coverage by surface functionalities, etc.
“with GNPs in DMEM medium at the concentration of 15 pM” was added in line 92.
“ Zeta potential of CTAB-GNPs and BSA-GNPs are 28.4±2.6 mV and -20.5±2.1 mV respectively, indicating positive CTAB and negative BSA on the surface of GNPs. ” was added in line 177.
3. Also, please define the controls used in Figures 2 and 3.
“when untreated cells used as control” was added in lines 208 and 237.
4. CTAB coated particles are referred as "toxic" although no data is provided. Could it be that CTAB detaches during incubations with cells and is responsible for the observed effects?
Line 224 was revised into “This could be attributed to the surfactant CTAB, which facilitates lysosome escape as reported in previous reports.”
Round 2
Reviewer 1 Report
The manuscript has been considerably revised and still needs some revisions given below:
Overall writing of the manuscript can be improved further.
Line 10: change ‘contribution’ to ‘study’
Line 15: change ‘bared’ to bare
Line 156: change to biocompatibility
Line 294: change to biocompatible
Line 308: change to ‘biocompatible coated GNPs’,
Author Response
Thank you for valuable advice, we have made the following corrections.
Line 10: change ‘contribution’ to ‘study’
“contribution” has been changed to “study”
Line 15: change ‘bared’ to bare
“bared” has been changed to “bare”
Line 156: change to biocompatibility
“its important role in encapsulation” has been changed to “biocompatibility”
Line 294: change to biocompatible
“biocompatibility” has been changed to “biocompatible”
Line 308: change to ‘biocompatible coated GNPs’
“Coating GNPs with biocompatible molecules” has been changed to “biocompatible coated GNPs”
Reviewer 2 Report
The manuscript is acceptable for publication.
Author Response
Thank you for your approval of the article, we made some further modifications.
Line 10“contribution” has been changed to “study”.
Line 15 “bared” has been changed to “bare”.
Line 15. “its important role in encapsulation” has been changed to “biocompatibility”.
Line 57 “misunderstandings” has been changed to “understandings”
Line 156 “poor biocompatible” has been changed to “poor biocompatibility”
Line 294“biocompatibility” has been changed to “biocompatible” .
Line 308 “Coating GNPs with biocompatible molecules” has been changed to “biocompatible coated GNPs”
Line 347 “application” has been changed to “applies”
Reviewer 3 Report
Thank you for the revised manuscript. Authors tried to address the reviewers concerns as outlined in the first report, however additional revision is warranted prior to publication.
I would strongly suggest to have a thorough proofreading, as there are several instances of English language misuse, e.g. line 57 "misunderstandings", line 156 "poor biocompatible", line 347 " applies".
Particle characterization by DLS requires a more extensive description of the modelling procedure, how a cylindrical shape particle size was obtained, given that the hydrodynamic radius implies a spherical particle shape. Also, details how the zeta potential was measured are lacking (solution pH, concentration, etc.)
Author Response
Thank you for valuable advice, we have made the following corrections.
Authors tried to address the reviewers concerns as outlined in the first report, however additional revision is warranted prior to publication.
I would strongly suggest to have a thorough proofreading, as there are several instances of English language misuse, e.g. line 57 "misunderstandings", line 156 "poor biocompatible", line 347 " applies".
Line 57 “misunderstandings” has been changed to “understandings”
Line 156 “poor biocompatible” has been changed to “poor biocompatibility”
Line 347 “application” has been changed to “applies”
Line 10“contribution” has been changed to “study”.
Line 15 “bared” has been changed to “bare”.
Line 15. “its important role in encapsulation” has been changed to “biocompatibility”.
Line 294“biocompatibility” has been changed to “biocompatible” .
Line 308 “Coating GNPs with biocompatible molecules” has been changed to “biocompatible coated GNPs”
Particle characterization by DLS requires a more extensive description of the modelling procedure, how a cylindrical shape particle size was obtained, given that the hydrodynamic radius implies a spherical particle shape. Also, details how the zeta potential was measured are lacking (solution pH, concentration, etc.)
Line 83 was changed to “37°C with GNPs in DMEM medium (pH 7.4) at the concentration of 15 pM.”
“Although hydrodynamic diameter deduced from Stokes-Einstein equation was not accurate when regard nano-rods as nano-spheres, the diffusion coefficient determined by dynamic light scattering (DLS) is still accurate. The particle size peak can be a signature for determining the nanorod aggregation formation [17].” and “17. Liu, H.; Nickisha, P.P.; Qun, H. Dynamic light scattering for gold nanorod size characterization and study of nanorod–protein interactions. Gold Bull. 2012, 45, 187–195”. were added.